# Scheduling of Generation Stations, OLTC Substation Transformers and VAR Sources for Sustainable Power System Operation Using SNS Optimizer

Ragab El-Sehiemy [1,*], Abdallah Elsayed [2], Abdullah Shaheen [3], Ehab Elattar [4] and Ahmed Ginidi [3]

1   Department of Electrical Engineering, Faculty of Engineering, Kafrelsheikh University,
    Kafrelsheikh 33516, Egypt
2   Department of Electrical Engineering, Faculty of Engineering, Damietta University, Damietta 34517, Egypt;
    am.elsherif@yahoo.com
3   Department of Electrical Engineering, Faculty of Engineering, Suez University, Suez 43533, Egypt;
    abdullahshaheen2015@gmail.com (A.S.); ahmed.ginidi@eng.suezuni.edu.eg (A.G.)
4   Department of Electrical Engineering, College of Engineering, Taif University, Taif 21944, Saudi Arabia;
    e.elattar@tu.edu.sa
*   Correspondence: elsehiemy@eng.kfs.edu.eg

**Abstract:** Typically, the main control on alternating current (AC) power systems is performed by the scheduling of rotary machines of synchronous generators and static machines of on-load tap changer (OLTC) transformers and volt-ampere reactive (VAR) sources. Large machines of synchronous generators can be managed by utilizing terminal voltage control when synchronized in parallel to the power system. These machines are typically terminal voltage regulated. In addition, substation on-load tap changer (OLTC) transformers improve system voltage management by controlling variable turn ratios that are adjusted in different levels known as taps along either the primary or secondary winding. Moreover, volt-ampere reactive (VAR) sources of static VAR compensators (SVCs), which are automated impedance devices connected to the AC power network, are designed for voltage regulation and system stabilization. In this paper, scheduling of these machines is coordinated for optimal power system operation (OPSO) using a recent algorithm of social network search optimizer (SNSO). The OPSO is performed by achieving many optimization targets of cost of fuel, power losses, and polluting emissions. The SNS is a recent optimizer that is inspired from users in social networks throughout the different moods of users such as imitation, conversation, disputation, and innovation mood. The SNSO is developed for handling the OPSO problem and applied on an IEEE standardized 57-bus power system and real Egyptian power system of the West Delta area. The developed SNSO is used in various assessments and quantitative analyses with various contemporary techniques. The simulated findings prove the developed SNSO's solution accuracy and resilience when compared to other relevant techniques in the literature.

**Keywords:** optimal power flow; social network search algorithm; electrical power grids; power losses; fuel costs; emissions

## 1. Introduction

The principal management of alternating current (AC) power systems is often operated by the scheduling of synchronous generator rotary machines and static machines of on-load tap changer (OLTC) transformers and volt-ampere reactive (VAR) sources. When large machines of synchronous generators are paralleled to the utility bus, terminal voltage control can be used to manage them. These machines are typically terminal voltage regulated. In addition, substation on-load tap changer (OLTC) transformers improve system voltage management by controlling variable turn ratios that are adjusted in different levels known as taps along either the primary or secondary winding. Moreover, volt-ampere eactive (VAR) sources of static VAR compensators (SVCs), which are automated impedance

devices connected to the AC power network, are designed for voltage regulation and system stabilization [1,2]. Real and reactive power management as a result of the installation of power generation at suitable buses can lead to a significant losses reduction and voltage regulation, especially in congested networks. Integration of renewable energy resources such as wind turbines and photovoltaic solar systems into the electricity network is currently a difficult task [3]. The power grid must fulfill two distinct necessities: maintaining a near real-time balance of generations and demands and adjusting managed machines to regulate active and reactive power flows via transmission network [4]. The electrical demands represent the aggregate of thousands of various consumers' power needs, extending from individual residences to huge commercial and industrial facilities [5,6].

The optimal power system operation (OPSO) is a non-linear, multi-model technique for power system control and operation. OPSO may be used to develop financial and safe operating conditions for power systems [7]. The OPSO can optimize one or more objectives such as fuel cost, emission of power system sources, and transmission losses [8]. These goals can be met while maintaining power flow balance and operating variables within their respective restrictions, such as voltage limits, line flow limits, valve constraints, and generator power [9].

Miscellaneous conventional mathematic approaches have been proposed to solve the OPSO such as semidefinite programming [10], non-linear programming [11], linear programming (LP) [12], Newton-based approach [13], interior-point methods (IPMs) [14], fuzzy linear programming [15], and sequential unconstrained minimization technique [16], and interior point method [17].

A variety of these methods can effectively enforce inequality constraints and have high convergence properties. However, because they rely on the initial settings, these traditional methods cannot produce the true optimal result and may become stuck in a local minimum. Furthermore, each technique must be represented using a specific variant(s) of OPSO, thus they cannot cope with integer and discrete variables seamlessly. As a result, developing metaheuristic approaches to overcome the aforementioned drawbacks is critical.

The finest solution of the OPSO can be determined by diverse augmentations of the algorithm techniques. An enhanced social spider optimization technique has been proposed, in ref. [18], by varying the movement strategy of male and female spiders to an appropriate ratio, to optimize fuel cost, emission, and losses independently. An enhanced NSGA-III with constraints handling, in environment selection operation and reducing selection efforts, has been illustrated to fuel cost, emission, and losses in [19]. In [20], a multi-objective backtracking search optimizer (MBSO) was demonstrated to formulate and solve the 30-bus, 57-bus, and 118-bus systems with objective functions fuel cost, voltage deviation, and power losses. An enhanced manta ray foraging technique (EMRFT), in ref. [21], has been characterized, in AC meshed power systems, for minimizing fuel cost, emission, and losses (with and without) voltage source converter (VSC) stations.

Experts are looking for ways to replace fossil energies with renewable generation in order to create ecologically friendly and emissions-free communities. In [22], PSO and GWO have been hybridized for solving OPSO issues, and they were combined with probabilistic photovoltaic and wind resources. In this study, wind speed distribution related to the wind turbines was presented through the Weibull probability distribution function [23]. Moreover, the produced power from the solar photovoltaic systems, which can be modeled as single and double-diode models [24–27], were presented through the lognormal probability distribution function. In [28], the OPSO has been discussed for the AC power flow tool where the DC flow tool has been investigated as well by linearizing the Ac variables in the system. In [29], a multi-period OPSO issue has been formulated considering the penetrations of variable renewable sources with uncertainties due to weather fluctuations.

In [30], an optimal generation scheduling has been presented including different renewable sources of photovoltaic, micro-turbine, wind, fuel cell and batteries. In this study, a beetle antenna search optimization has been employed considering hourly loadings in

real time. In [31], the equilibrium optimizer technique (EO) was applied for the OPSO issue incorporating different renewable sources by formulating their uncertainties via probability density functions in order to expect their produced power. In this work, many objective functions were considered and handled using the weight factors. In [32], a technique for effectively distributing various kinds of renewable resources in the distribution network has been developed but the reduction of yearly energy losses has been framed as a single objective optimization framework. In [33], the EO technique was applied for integrating the photovoltaic distributed generations and batteries in distribution systems. In this study, many objective functions were taken into account of improving the reliability, minimizing the investment costs, reducing the power losses, and minimizing the environmental emissions but they were handled in a single objective model.

Despite these performed applications for solving the OPSO, the simplifications by ignoring the reactive power injections from capacitive sources and transformer tap settings potentially lead to inaccurate results. In this paper, a scheduling of synchronous generator rotary machines and static machines of on-load tap changer (OLTC) transformers and volt-ampere reactive (VAR) sources is coordinated for optimal power system operation (OPSO) using a recent algorithm of social network search Optimizer (SNSO). The OPSO is performed by achieving many optimization targets of cost of fuel, power losses, and polluting emissions. The SNSO is a recent optimizer that is inspired by users in social networks throughout the different moods of users such as imitation, conversation, disputation, and innovation mood [34]. The SNSO is developed for handling the OPSO problem and applied on an IEEE standardized 57-bus power system and real Egyptian power system of the West Delta area. The developed SNSO is used in various assessments and quantitative analyses with various contemporary techniques. The key contributions of this paper are as follows:

- The developed SNSO has been employed to minimize the objective functions of fuel costs, losses, and emissions in electrical power networks and applied on the standardized network of IEEE 57-bus and a practical Egyptian network of WDA.
- The developed SNSO provides better performance than various recent techniques.
- Significant stability is demonstrated for the developed SNSO for solving the OPSO in electrical power networks.
- A validation assessment is conducted for the rotary and static machines of the IEEE 57-bus and WDA power systems.
- High validation is illustrated based on the SNSO for the optimal scheduling of synchronous generator rotary machines and static machines of on-load tap changer (OLTC) transformers and Volt-Ampere Reactive (VAR) sources.

The remaining sections of this paper are considered as follows: the OPSO formulation is established in Section 2, while the developed SNSO for OPSO is manifested in Section 3. Moreover, the simulation results and discussion are illustrated in Section 4, and conclusion remarks are given in Section 5.

## 2. Problem Formulation

In OPSO, the independent/decision and the dependent variables are manifested. The active power outputs of the generators and the reactive power injection of switched capacitors and reactors are represented by ($Pg_1$, $Pg_2$, ..., $Pg_{Ng}$) and ($Qc_1$, $Qc_2$, ..., $Qc_{Nq}$), respectively. The generator voltages and the transformer tap settings are denoted by ($Vg_1$, $Vg_2$, ..., $Vg_{Ng}$) and ($Tap_1$, $Tap_2$, ... ..., $Tap_{Nt}$), respectively, Where the number of generators, the number of on-load tap changing transformers, and the number of the VAR sources are demonstrated, respectively, by Ng, Nt, and Nq. The dependent variables include load bus voltage magnitudes, generator reactive power outputs of the generators and transmission line loadings, which are demonstrated by ($VL_1$, ..., $VL_{NPQ}$), ($Qg_1$, $Qg_2$, ..., $Qg_{Ng}$), and ($SF_1$, ..., $SF_{NF}$), Where the number of load buses and the transmission lines are illustrated by NPQ and NF, respectively. This issue can be formulated mathematically as follows:

$$\text{Min F} = \{OV_1(u, v), OV_2(u, v) \dots, OV_m(u, v)\} \text{ subjected to}: g(u, v) = 0 \text{ and } h(u, v) \leq 0 \tag{1}$$

where F illustrates the considered vector of diverse m objectives; u and v are the decision and the dependent variables, respectively.

### 2.1. Problem Objectives

The first objective is the fuel generation costs ($OV_1$) in USD/h as depicted in (2):

$$OV_1 = \sum_{i=1}^{Ng} \left( a_i Pg_i^2 + b_i Pg_i + c_i \right) \tag{2}$$

where $Pg_i$ refers to the active output power in MW of each generator i; $a_i$, $b_i$, and $c_i$ represent the relevant cost coefficients of each generator i.

The second objective function includes the total ton/h emissions ($OV_2$) of the atmospheric pollutants which are expressed as in Equation (3):

$$OV_2 = \sum_{i=1}^{Ng} \left( (\gamma_i Pg_i^2 + \beta_i Pg_i + \alpha_i)/100 + \zeta_i e^{\lambda_i Pg_i} \right) \tag{3}$$

where $\gamma_i$, $\beta_i$, $\alpha_i$, $\zeta_i$ and $\lambda_i$ denote the atmospheric pollutants emission coefficients of generator i.

The third objective involves minimizing the total power losses of the transmission network as expressed [35]:

$$OV_3 = \sum_{i=1}^{Nb} \sum_{j=1}^{Nb} G_{ij}(V_i^2 + V_j^2 - 2(V_i V_j \cos \theta_{ij})) \tag{4}$$

where $G_{ij}$ is the conductance of each line between bus i and j; Nb is the total number of buses; V indicates voltage and $\theta$ refers to the phase angle.

### 2.2. System Constraints

The equality constraints are manifested by the load flow balance equations as depicted in the following equation:

$$Pg_i - PL_i - V_i \sum_{j=1}^{Nb} V_j \left( G_{ij} \cos \theta_{ij} + B_{ij} \sin \theta_{ij} \right) = 0, \ i = 1, \dots, Nb \tag{5}$$

$$Qg_i - QL_i + Qc_i - V_i \sum_{j=1}^{Nb} V_j \left( G_{ij} \sin \theta_{ij} - B_{ij} \cos \theta_{ij} \right) = 0, \ i = 1, 2, \dots, Nb \tag{6}$$

where PL and QL indicate both the active and reactive power demand, respectively. Moreover, $G_{ij}$ and $B_{ij}$ define the mutual conductance and susceptance between bus i and j, respectively.

Furthermore, the operational variables and their corresponding constraints, denoted by the superscripts "min" and "max" limits, are formulated as follows:

$$Pg_i^{min} \leq Pg_i \leq Pg_i^{max}, \ i = 1, 2, \dots, Ng \tag{7}$$

$$Vg_i^{min} \leq Vg_i \leq Vg_i^{max}, \ i = 1, 2, \dots, Ng \tag{8}$$

$$Qg_i^{min} \leq Qg_i \leq Qg_i^{max}, \ i = 1, 2, \dots, Ng \tag{9}$$

$$Tap_k^{min} \leq Tap_k \leq Tap_k^{max}, \ k = 1, 2, \dots, Nt \tag{10}$$

$$Qc_q^{max} \leq Qc_q \leq Qc_q^{max}, \ q = 1, 2, \dots, Nq \tag{11}$$

$$VL_i^{min} \leq VL_i \leq VL_i^{max}, \ i = 1, \ 2, \ \ldots, \ NPQ \tag{12}$$

$$|S_f| \leq S_f^{max}, \ f = 1, \ 2, \ \ldots, \ Nf \tag{13}$$

where $S_f$ is the transmission line flows in line f; $VL_i$ is the voltage of load bus i; NPQ and Nf are the total number of load buses and system lines, respectively.

## 3. Developed SNSO for OPSO in Power Systems

### 3.1. SNSO

Individuals in social networking sites drive the social network search optimizer (SNSO), which seeks to be appealing across different user moods such as imitation, conversation, disputation, and innovation [34]. These moods are methods for expressing people's fresh thoughts on a new occurrence. Other users' points of view are appealing in the imitation mood, and users commonly strive to imitate one another in expressing their ideas. Users with the conversation mood can communicate with one another and benefit from one another's viewpoints. Others in a dispute mood can engage in a debate with a group of people and discuss their points of view. Users in the innovation mood post a topic on social media, usually based on their new opinions and experiences. The mathematical modeling and explanation of these (moods) are shown below:

### 3.1.1. Imitations

If a new event with an exciting concept is launched in this mood, users may try to imitate the renowned individuals who share their thoughts by writing a conversation about this event. This mental state can be represented quantitatively as follows:

$$X_{i,new} = X_j + rand(-1,1) \times rand(0,1) \times (X_i - X_j) \tag{14}$$

where $X_i$ $X_j$ exemplifies the vector, which is selected randomly, of the *j*th user's view (position) and the vector of the *i*th user's view, respectively, and $i \neq j$. In addition to this, the two terms rand (0, 1) and rand (−1, 1) indicate two random vectors in intervals [0, 1] and [−1, 1], respectively.

### 3.1.2. Conversations

People in this mood can improve their understanding of an event by learning from one another and exploring ideas about the event from many viewpoints, helping them to develop a fresh perspective on the event. This mental state can be represented quantitatively as follows:

$$X_{i,new} = X_k + rand(0,1) \times [sign(f_i - f_j) \times (X_i - X_j)] \tag{15}$$

where $X_k$ $_k$ illustrates the vector of the issue which is randomly selected to speak about it. Moreover, *R* represents the impact of chat. This impact depends on the opinions' differences and characterizes the change in their views about the event ($X_k$), while *D* illustrates the difference among the beliefs of users. In addition, $X_j$ displays the vector of a randomly selected user's belief for a talk and $X_i$ displays the vector of view of the *i*th user and $i \neq j \neq k$. Additionally, the term (*sign*) illustrates the sign function, while the term (*sign(f_i-f_j)*) establishes a comparison between $f_i$ and $f_j$ which illustrates the moving direction of $X_k$.

It can be noted that the user's viewpoint about the event changes because of conversations with the *j*th user, where the developed opinion represents a new belief to share with others. Adjusting the user's belief about the events is considered as the replacement of the events.

### 3.1.3. Disputations

In this mood, people can defend their ideas by describing them in comments or establishing groups' sections; nevertheless, they can be influenced by other commentators

or members of a virtual group that has been formed to discuss a point of view on a certain issue. The new impacted view can be expressed mathematically as follows:

$$X_{i,new} = X_i + rand(0,1) \times \left( \frac{\sum_{m=1}^{N_r} X_m}{N_r} - ((1 + round(rand(0,1))) \times X_i) \right) \quad (16)$$

where *M* represents the mean of views of friends in the group or commenters, while the term (*AF*) expresses the Admission Factor that illustrates the assertion from users on their opinion in discussions and represented as an integer number of 1 or 2. The symbol (*round*) rounds the real input to the adjacent integer number, whereas the symbol ($N_r$) represents the group size or commenters and represents a random number between 1 and *Nuser* (the number of users of the network).

### 3.1.4. Innovations

Users can express their own opinions and feelings regarding a certain event in unique and inventive ways while in this mindset. As a result, a new concept will be generated, and the new impacted viewpoint may be mathematically expressed as follows:

$$X_{i,new}^d = tX_j^d + (1-t) \times (LB^d + rand(0,1) \times (UB^d - LB^d))$$
$$t = rand(0,1) \quad (17)$$

where the symbol (*d*) illustrates the $d^{\text{th}}$ variable in the interval [1,*D*] which is selected randomly, and (*D*) manifests the problem variables' number, whereas the two variables ($rand_1$) and ($rand_2$) are random numbers with interval of [0, 1]. In addition to that, $ub_d$ and $lb_d$ are upper and lower values of the *d*th variable, whilst $n_{new}^d$ signifies the new thought about the $d^{\text{th}}$ dimension of the problem. The variable $x_j^d$ characterizes the existing thought about $d^{\text{th}}$ variable produced by another user $j^{\text{th}}$ user ($i \neq j$) and *i*th user requires to adjust it due to new thought ($n_{new}^d$). As a result, the new view $X_{inew}^d$ about the $d^{\text{th}}$ dimension will be established. $X_{inew}^d$ is an interpolation about the existing thought ($n_j^d$) and the new thought ($n_{new}^d$).

As a result, a change in one dimension ($X_{inew}^d$) creates a general shift in the fundamental notion and may be seen as a new point of view to be conveyed. As a result, this process may be mathematically expressed as follows:

$$X_{i,new} = [x_1, x_2, x_3, \ldots \ldots, x_{i,new}^d, \ldots x_D] \quad (18)$$

It is illustrated that $X_{i,new}$ elaborates a new perception into the event in accordance with the $d^{\text{th}}$ viewpoint and substituted with the existing view $x_i^d$ as depicted in Equation (18).

### 3.1.5. Rules Related to Network

Each social network defines a set of roles for its users, and these roles are regarded by all users in shared perspectives. The following factors are used to limit the users' perspectives:

$$x_i = \min(x_i, UB_i) \& x_i = \max(x_i, LB_i), i = 1, 2, \ldots \ldots D \quad (19)$$

where $x_i$ manifests the $i^{\text{th}}$ variable of (new idea) $X_{inew}$, while $LB_i$ and $ub_i$ depicts the $i^{\text{th}}$ component of *UB* and *LB* of problem.

### 3.1.6. Rules for Publishing

The method of this algorithm is produced by various moods, where each user's viewpoint is altered, and fresh views are utilized based on their merit. To demonstrate, if the new idea is superior to the existing one, it will be approved. As a result, the value of a

new concept may be determined by the objective function of $X_{i,new}$, which can be calculated analytically and compared to the value of an existing thought $(X_i)$ as follows:

$$X_i = \begin{cases} X_i & f(X_{i,new}) > f(X_i) \\ X_{i,new} & f(X_{i,new}) < f(X_i) \end{cases} \tag{20}$$

To execute the method, the maximum number of iterations (MaxIter), the number of users (N), and variable limitations must be manifested, where the starting view for each user may be determined as in Equation (21):

$$X_0 = LB + rand(0,1) \times (UB - LB) \tag{21}$$

where $X_o$ illustrates the primitive view vector for each user, whilst *UB* and *LB* represent upper and lower vectors of the variables, respectively. Later, the objective function for each user's viewpoints is computed. Figure 1 describes the main steps of the SNST. From that figure, the random process governs the selection of one mood from the imitations, conversations, disputations, and innovations modes that are described in Equations (14)–(17), respectively. In these moods, there is no specified parameters are used as elements of the vectors described but they are randomly created and updated. its random updating mechanism shows further advantage since the SNSO algorithm isn't dependent on specified parameters which makes it very sensitive to its choices.

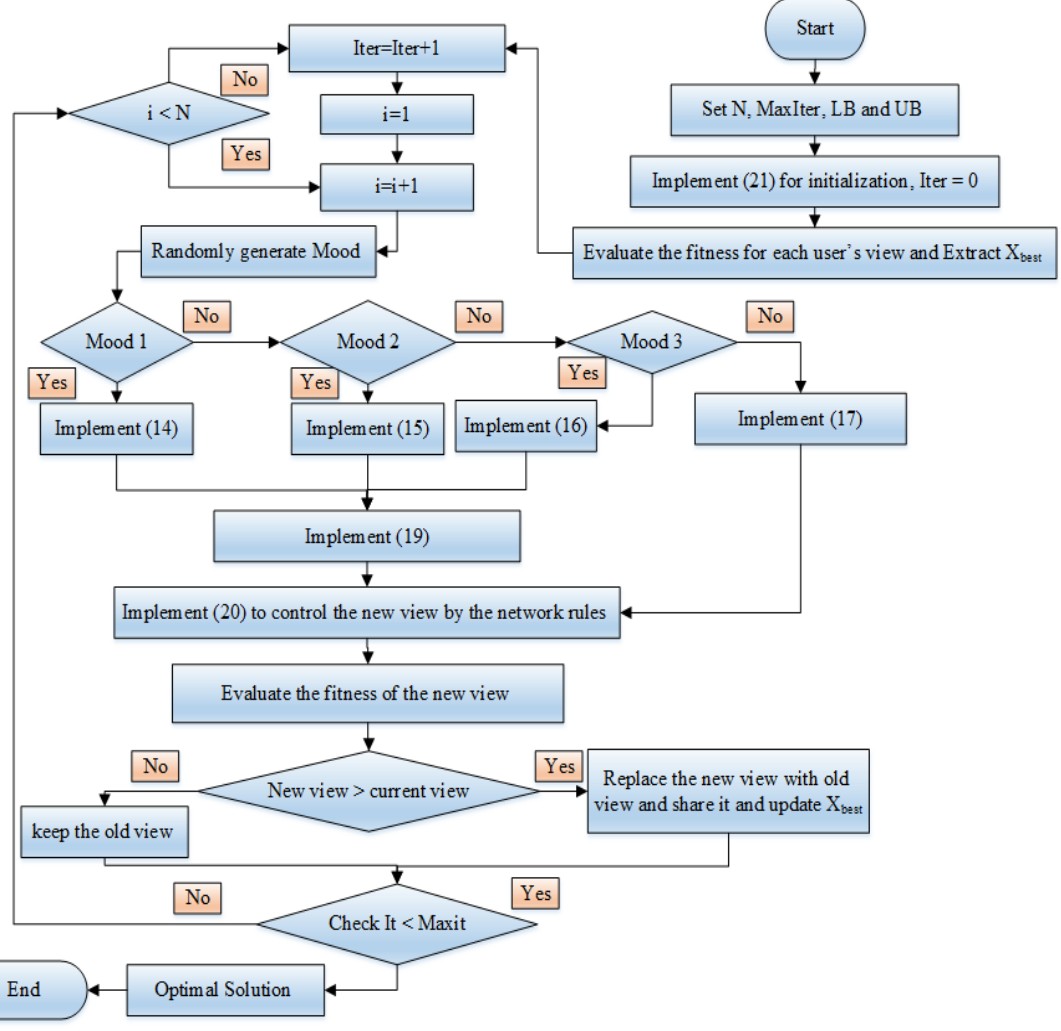

**Figure 1.** SNSO Flowchart.

### 3.2. Developed SNSO for OPSO

When dealing with the previously described OPSO challenge, the equality and inequality requirements are taken into account. The Newton–Raphson (NR) technique is used to fulfill the equality criteria that characterize load flow balancing equations. The NR technique maintains the balancing requirements of (5) and (6) because it describes the service's steady-state for power network engineers, it is included in this section. As a result, the NR approach exemplifies an important platform for showing three-phase circuits and is employed by MATPOWER [36]. Two operational constraints reflect any other of the constraints, that are decision and dependent variable constraints.

#### 3.2.1. SNSO Development for Including Opertaional Limits of Decision Variables

The first category (decision variables) continues to achieve their limits, and if either of them exceeds assessments, they are randomly recreated within the suitable bounds. Therefore, Equations (7)–(11) can be re-formulated as follows:

$$Pg_k = \begin{cases} Pg_k^{min} & \text{if } Pg_k \leq Pg_k^{min} \\ Pg_k^{max} & \text{if } Pg_k \geq Pg_k^{max} \end{cases}, \quad k = 1, 2, \ldots, Ng \tag{22}$$

$$Vg_k = \begin{cases} Vg_k^{min} & \text{if } Vg_k \leq Vg_k^{min} \\ Vg_k^{max} & \text{if } Vg_k \geq Vg_k^{max} \end{cases}, \quad k = 1, 2, \ldots, Ng \tag{23}$$

$$Qg_k = \begin{cases} Qg_k^{min} & \text{if } Qg_k \leq Qg_k^{min} \\ Qg_k^{max} & \text{if } Qg_k \geq Qg_k^{max} \end{cases}, \quad k = 1, 2, \ldots, Ng \tag{24}$$

$$Tap_{Tr} = \begin{cases} Tap_{Tr}^{min} & \text{if } Tap_{Tr} \leq Tap_{Tr}^{min} \\ Tap_{Tr}^{max} & \text{if } Tap_{Tr} \geq Tap_{Tr}^{max} \end{cases}, \quad Tr = 1, 2, \ldots, Nt \tag{25}$$

$$Qc_{VAR} = \begin{cases} Qc_{VAR}^{min} & \text{if } Qc_{VAR} \leq Qc_{VAR}^{min} \\ Qc_{VAR}^{max} & \text{if } Qc_{VAR} \geq Qc_{VAR}^{max} \end{cases}, \quad VAR = 1, 2, \ldots, Nq \tag{26}$$

#### 3.2.2. SNSO Development for Including Opertaional Limits of Dependent Variables

Furthermore, the targeted cost function extends and penalizes the constraints of the second category (dependent variables). As a result, if the solution related to the view of users violates any of the corresponding limitations, it will be rejected in the following iteration. The considered objective goal (OJ) can be defined using these principles:

$$OV = OV_i + Pen_v \sum_{NPQ} \Delta V_L^2 + Pen_Q \sum_{Nq} \Delta Q_G^2 + Pen_{SF} \sum_{N_f} \Delta S_F^2, \quad i = 1, \ldots \ldots m \tag{27}$$

whereas, $\Delta V_L$, $\Delta Q_G$, and $\Delta S_F$ are represented as:

$$\Delta V_L = \begin{cases} V_L^{min} - V_L & \text{if } V_L < V_L^{min} \\ V_L^{max} - V_L & \text{if } V_L > V_L^{max} \end{cases} \tag{28}$$

$$\Delta Q_G = \begin{cases} Q_G^{min} - Q_G & \text{if } Q_G < Q_G^{min} \\ Q_G^{max} - Q_G & \text{if } Q_G > Q_G^{max} \end{cases} \tag{29}$$

$$\Delta S_F = S_F^{max} - S_F \text{ if } S_F > S_F^{max} \tag{30}$$

where $OJ_i$ refers to each objective goal of the m goals; $Pen_v$, $Pen_q$ and $Pen_f$, are the penalty factors for the violations in load voltages, reactive outputs from generators and line power flows. Figure 2 describes the main steps of the developed solution based SNSO for OPSO in electrical power systems.

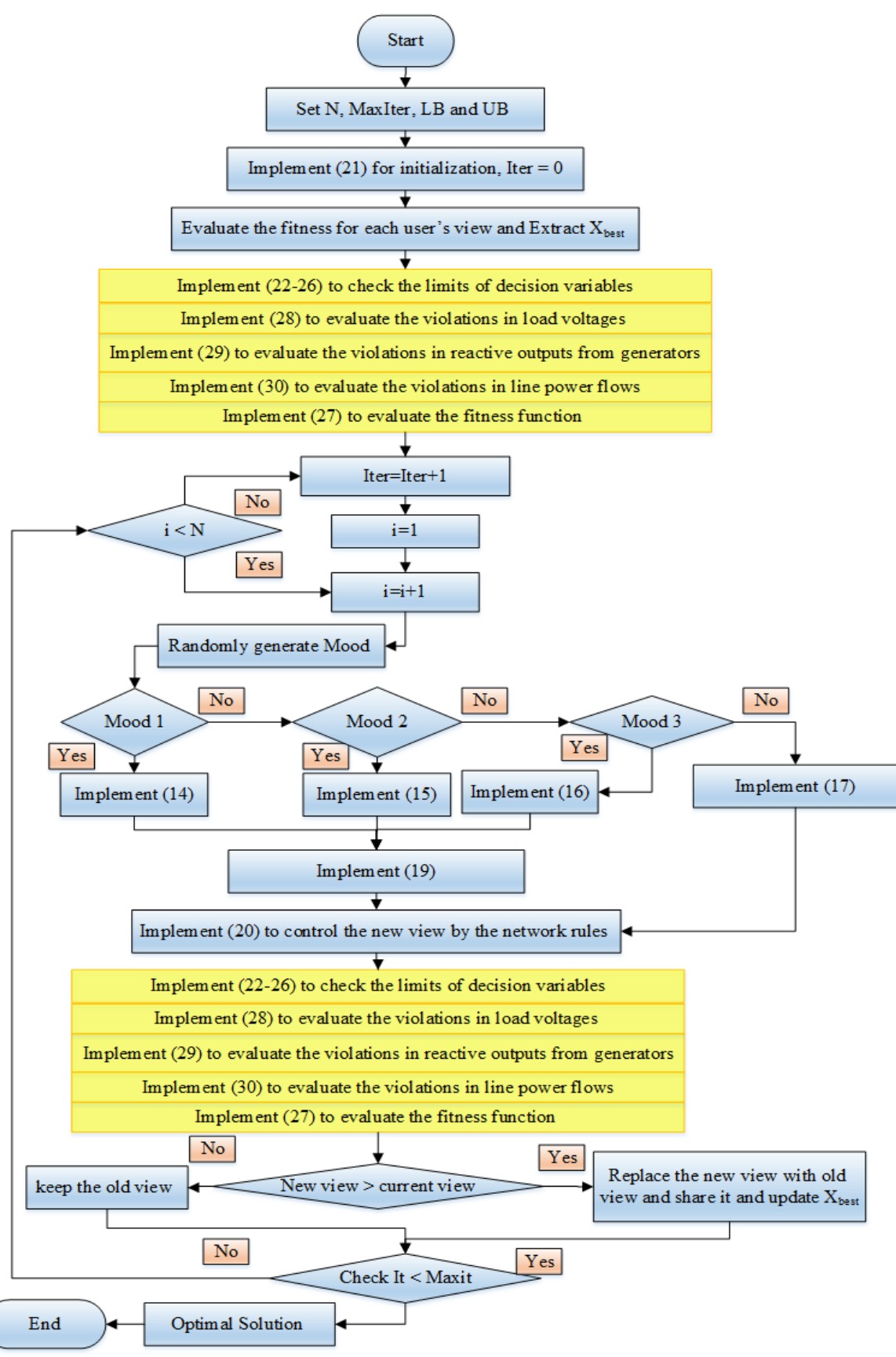

**Figure 2.** SNSO for solving the OPSO in electrical power systems.

## 4. Simulation Results

The designed SNSO is employed on two test power systems. The first is a typical IEEE 57-bus power system, and the second is an actual Egyptian power system known as the West Delta area (WDA) power system. Thirty simulated tests are performed for the created SNSO, with a maximum iteration of 300 and a user view of 25 members. As previously stated, the initial power system consists of 57 buses, 80 lines, 17 on-load tap changing transformers, 7 generators on buses 1, 2, 3, 6, 8, 9, and 12, and three capacitive sources on buses 18, 25, and 53. The statistics for buses, the minimum and maximum reactive power generation limitations, and transmission lines are extracted from [37]. Table 1 illustrates the cost and emission coefficients for IEEE 57-bus power system.

**Table 1.** Cost and emission coefficients for IEEE 57-bus power system [36].

| Generator | $a_i$ | $b_i$ | $c_i$ | $\alpha_i$ | $\beta_i$ | $\gamma_i$ | $\zeta_i$ | $\lambda_i$ |
|---|---|---|---|---|---|---|---|---|
| 1 | $7.76 \times 10^{-2}$ | 20 | 0 | 6 | $-5$ | 4 | $2 \times 10^{-5}$ | 0.5 |
| 2 | $1 \times 10^{-2}$ | 40 | 0 | 5 | $-6$ | 3 | $5 \times 10^{-5}$ | 1.5 |
| 3 | $25 \times 10^{-2}$ | 20 | 0 | 4 | $-5$ | 4 | $1 \times 10^{-5}$ | 1 |
| 6 | $1 \times 10^{-2}$ | 40 | 0 | 3.5 | $-3$ | 3.5 | $2 \times 10^{-5}$ | 0.5 |
| 8 | $2.22 \times 10^{-2}$ | 20 | 0 | 4.5 | $-5$ | 5 | $4 \times 10^{-5}$ | 2 |
| 9 | $1 \times 10^{-2}$ | 40 | 0 | 5 | $-4$ | 4.5 | $1 \times 10^{-5}$ | 2 |
| 12 | $3.23 \times 10^{-2}$ | 20 | 0 | 5 | $-5$ | 6 | $1 \times 10^{-5}$ | 1.5 |

The configuration of the real power system, which includes 52 buses [15]. Data of lines and buses are gathered from [38,39]. The maximum and minimum generator voltages are 1.06 and 0.94 p.u., respectively. MatlabR2017b is used to run the simulations, which are run on a CPU (2.5 GHz) Intel(R)-Core (TM) i7-7200U with 8 GB of RAM.

A tap changer mechanism (TCM) is a device in transformers that enables the selection of varied turn ratios in discrete stages. This is accomplished by connection to a series of entry points termed as taps located along the secondary or primary windings. TCMs come in two major kinds on-load and on-load mechanisms. The first TCM must be turned off first before the turns ratio is altered while the second one may change the ratio while servicing. The selection of tap points can be performed automatically, as is typically the case with on-load TCM, or manually, as no-load TCM. TCMs are commonly mounted on the side of high-voltage windings in power systems for the convenience of accessing and to decrease the current burden while servicing. A TCM controls the turn ratio in discrete steps. It operates with some steps in the positive and negative direction that provides $\pm 10\%$ turn ratio variation. Therefore, the maximum and minimum limits of the tap settings are 1.1 and 0.90 p.u., accordingly.

### 4.1. The First System Results

The following three scenarios are investigated as follows:

- Scenario 1: $OV_1$ Minimization of Fuel Generation Costs (FGC) that manifested in Equation (2)
- Scenario 2: $OV_2$ Minimization of Produced Emissions (PE) that manifested in Equation (3)
- Scenario 3: $OV_3$ Minimization of Overall Power Loss (OPL) that manifested in Equation (4)

#### 4.1.1. FGCs Minimizing (Scenario 1)

For this scenario, the developed SNSO is applied, where their obtained outputs are recorded in Table 2. Added to that, Figure 3 illustrates the convergence feature of the developed SNSO for Scenario 1. As shown, the developed SNSO minimizes from 41,685.5 USD/h at the initial scenario to 51,345 USD/h. This reduction represents a percentage of 18.81%.

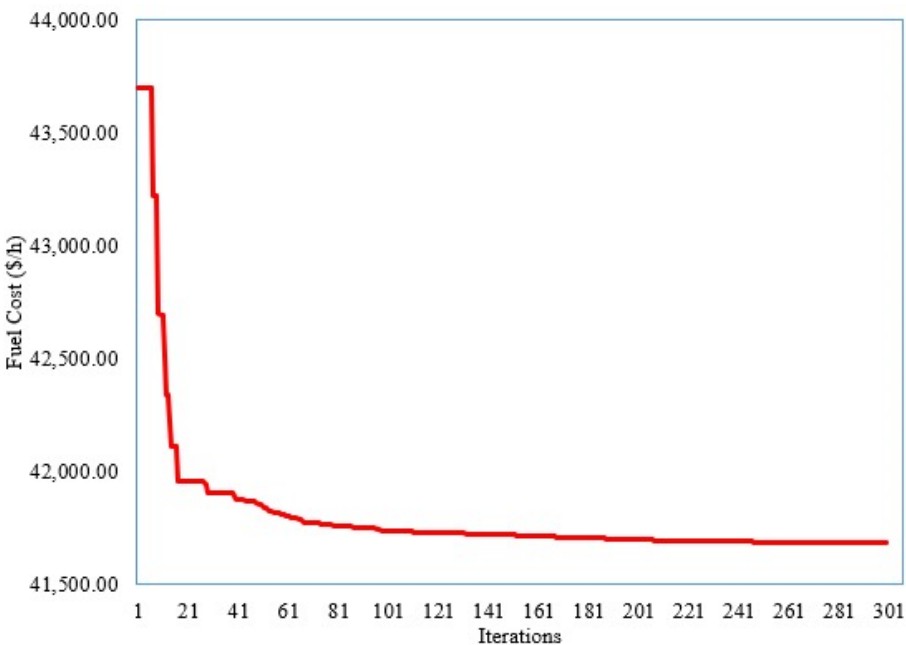

**Figure 3.** Convergence feature of developed SNSO for Scenario 1.

Table 3 further compares the outcomes of minimizing the FGCs (Scenario 1) with numerous alternative techniques which are real coded biogeography-based optimization [40], social spider optimization [18], enhanced social spider optimization [18], salp swarm optimizer [41], bat Search algorithm [42], electromagnetic field optimization [43], modified imperialist competitive algorithm [44], improved salp swarm optimizer [45], genetic algorithm [19], improved genetic algorithm [19] and differential search algorithm [46]. As shown, the developed SNSO demonstrates the best performance over the others since it obtains the minimum FGCs of 41,685.5 USD/h among other techniques. The whole solutions of the developed SNSO algorithm, enhanced social spider optimizer [18], improved salp swarm optimizer [45], salp swarm optimizer [41], and sli me mould algorithm [47] are revealed in Table 2. The solutions are analyzed and by checking their performance, Figure 4 describes the inequality constraints related to MVAr outputs of the generators. Despite the better performance of the developed SNSO, some techniques provide lesser FGCs values. Enhanced social spider optimization [18], salp swarm optimizer [41] and improved salp swarm optimizer [45] achieved FGCs of 41,665.540, 41,672.3, and 41,675.02 USD/h. This figure demonstrates that some violations of the inequality constraints related to MVAr outputs of the generators which declares the inadequacy of the acquired operating conditions of enhanced social spider optimization [18], salp swarm optimizer [41], and improved salp swarm optimizer [45].

**Table 2.** Optimal results using the developed SNSO for Scenario 1.

| Variables | Initial Scenario | Developed SNSO | Slime Mould Algorithm [47] | Enhanced Social Spider Optimizer [18] | Improved Salp Swarm Optimizer [45] | Salp Swarm Optimizer [41] |
|---|---|---|---|---|---|---|
| $Vg_1$ | 1.010 | 1.0364 | 1.050 | 1.046 | 1.062 | 0.925 |
| $Vg_2$ | 1.010 | 1.0342 | 1.0497 | 1.045 | 1.066 | 0.914 |
| $Vg_3$ | 1.010 | 1.0285 | 1.0491 | 1.043 | 1.054 | 1.082 |
| $Vg_6$ | 1.010 | 1.032 | 1.0638 | 1.056 | 1.060 | 0.960 |
| $Vg_8$ | 1.010 | 1.039 | 1.847 | 1.068 | 1.074 | 0.903 |
| $Vg_9$ | 1.010 | 1.016 | 1.050 | 1.037 | 1.062 | 1.035 |
| $Vg_{12}$ | 1.010 | 1.019 | 1.045 | 1.033 | 1.048 | 1.083 |
| $Tap_{4-18}$ | 0.970 | 0.907 | 1.048 | 5.335 | 1.030 | 0.950 |
| $Tap_{4-18}$ | 0.978 | 1.009 | 0.916 | 5.900 | 0.989 | 0.913 |
| $Tap_{21-20}$ | 1.043 | 1.032 | 1.015 | 6.299 | 1.013 | 0.988 |
| $Tap_{24-25}$ | 1.000 | 1.000 | 0.913 | 0.904 | 0.958 | 0.950 |
| $Tap_{24-25}$ | 1.000 | 1.047 | 0.927 | 0.963 | 0.999 | 1.088 |

**Table 2.** *Cont.*

| Variables | Initial Scenario | Developed SNSO | Slime Mould Algorithm [47] | Enhanced Social Spider Optimizer [18] | Improved Salp Swarm Optimizer [45] | Salp Swarm Optimizer [41] |
|---|---|---|---|---|---|---|
| Tap $_{24-26}$ | 1.043 | 1.010 | 1.025 | 1.009 | 1.017 | 1.013 |
| Tap $_{7-29}$ | 0.967 | 0.932 | 0.991 | 0.971 | 0.995 | 0.900 |
| Tap $_{34-32}$ | 0.975 | 0.958 | 0.927 | 0.957 | 0.947 | 0.988 |
| Tap $_{11-41}$ | 0.955 | 0.904 | 0.906 | 1.024 | 1.005 | 0.963 |
| Tap $_{15-45}$ | 0.955 | 0.923 | 0.971 | 0.950 | 0.978 | 0.913 |
| Tap $_{14-46}$ | 0.900 | 0.916 | 0.967 | 0.946 | 0.975 | 0.913 |
| Tap $_{10-51}$ | 0.930 | 0.918 | 0.978 | 0.900 | 0.991 | 0.913 |
| Tap $_{13-49}$ | 0.895 | 0.904 | 0.932 | 0.928 | 0.946 | 0.900 |
| Tap $_{11-43}$ | 0.958 | 0.904 | 0.978 | 0.918 | 0.971 | 0.900 |
| Tap $_{40-56}$ | 0.958 | 1.001 | 1.003 | 0.923 | 1.010 | 1.000 |
| Tap $_{39-57}$ | 0.980 | 0.976 | 0.971 | 0.900 | 0.999 | 1.050 |
| Tap $_{9-55}$ | 0.940 | 0.932 | 0.991 | 0.920 | 0.998 | 0.900 |
| Qc $_{18}$ | 10.000 | 11.169 | 17.167 | 1.004 | 13.732 | 8.817 |
| Qc $_{25}$ | 5.900 | 15.504 | 12.982 | 0.977 | 9.874 | 13.517 |
| Qc $_{53}$ | 6.300 | 11.427 | 21.653 | 0.943 | 11.595 | 5.023 |
| Pg $_1$ | 478.635 | 142.612 | 147.486 | 140.821 | 144.026 | 148.346 |
| Pg $_2$ | 0.000 | 89.830 | 87.582 | 88.426 | 96.229 | 64.555 |
| Pg $_3$ | 40.000 | 44.610 | 45.050 | 45.096 | 46.325 | 46.574 |
| Pg $_6$ | 0.000 | 66.728 | 72.342 | 72.084 | 81.514 | 78.152 |
| Pg $_8$ | 450.000 | 462.504 | 461.412 | 459.918 | 455.150 | 466.866 |
| Pg $_9$ | 0.000 | 93.781 | 97.109 | 96.532 | 74.029 | 89.775 |
| Pg $_{12}$ | 310.000 | 365.867 | 360.375 | 360.783 | 368.030 | 372.973 |
| Cost_Pg | 51,345 | 41,685.500 | 41,697.119 | 41,665.540 | 41,675.020 | 41,672.300 |
| Losses | 27.835 | 15.132 | 15.556 | 16.542 | 14.529 | 16.441 |

**Table 3.** Comparisons for Scenario 1.

| Technique | FGCs (USD/h) |
|---|---|
| Developed SNSO | 41,685.500 |
| Real Coded Biogeography-Based Optimization [40] | 41,686.000 |
| Social Spider Optimization [18] | 41,734.337 |
| Enhanced Social Spider Optimization [18] * | 41,665.540 |
| Bat Search Algorithm [42] | 41,686.820 |
| differential search algorithm [46] | 41,686.820 |
| Electromagnetic Field Optimization [43] | 41,706.117 |
| Salp Swarm Optimizer [41] * | 41,672.300 |
| Modified Imperialist Competitive Algorithm [44] | 41,738.440 |
| Genetic Algorithm [19] | 41,796.840 |
| Improved Genetic Algorithm [19] | 41,719.890 |
| Improved Salp Swarm Optimizer [45] * | 41,675.020 |

* indicates violated inequality constraints.

### 4.1.2. PEs Minimizing (Scenario 2)

As demonstrated in Table 4, the designed SNSO minimizes the PEs in the third scenario. The resulting PE value is 1.038 ton/h, as indicated in the table. Furthermore, Figure 5 shows the convergence features of the produced SNSO for Scenario 3. Table 5 shows how it compares to other metaheuristic optimization approaches. As demonstrated, the developed SNSO meets the minimal PE target of 1.038 ton/h. In terms of the minimum ability, it beats the other metaheuristics of the improved genetic algorithm [19], social spider optimization [18], Improved genetic algorithm [19], enhanced social spider optimization [18], teaching-learning based optimization [37], and modified imperialist competitive algorithm [44] in minimizing the PEs.

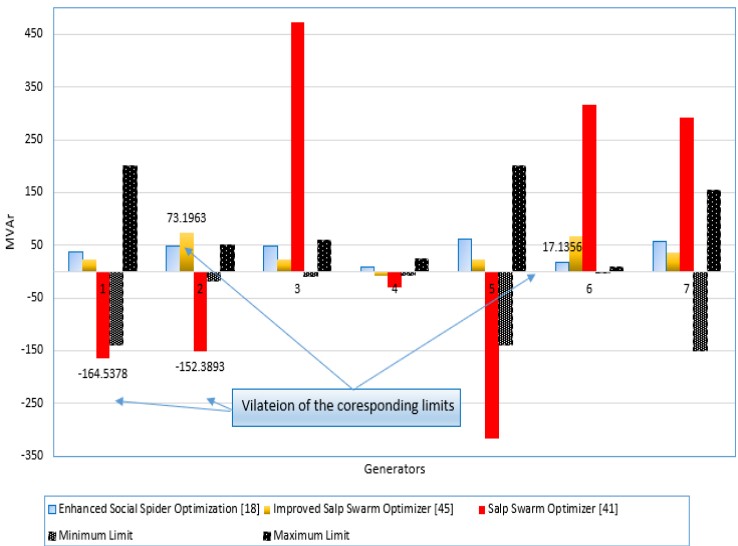

**Figure 4.** Violation of inequality constraints related to MVAr outputs of the generators.

**Table 4.** Optimal results using the developed SNSO for Scenario 2.

| Variables | Initial Scenario | Scenario 2 (PEs (Ton/h)) |
|---|---|---|
| $Vg_1$ | 1.010 | 1.060 |
| $Vg_2$ | 1.010 | 1.059 |
| $Vg_3$ | 1.010 | 1.060 |
| $Vg_6$ | 1.010 | 1.059 |
| $Vg_8$ | 1.010 | 1.060 |
| $Vg_9$ | 1.010 | 1.042 |
| $Vg_{12}$ | 1.010 | 1.050 |
| $Tap_{4-18}$ | 0.970 | 1.013 |
| $Tap_{4-18}$ | 0.978 | 1.08 |
| $Tap_{21-20}$ | 1.043 | 1.072 |
| $Tap_{24-25}$ | 1.000 | 1.092 |
| $Tap_{24-25}$ | 1.000 | 0.932 |
| $Tap_{24-26}$ | 1.043 | 1.004 |
| $Tap_{7-29}$ | 0.967 | 0.975 |
| $Tap_{34-32}$ | 0.975 | 0.937 |
| $Tap_{11-41}$ | 0.955 | 0.909 |
| $Tap_{15-45}$ | 0.955 | 0.956 |
| $Tap_{14-46}$ | 0.900 | 0.956 |
| $Tap_{10-51}$ | 0.930 | 0.986 |
| $Tap_{13-49}$ | 0.895 | 0.940 |
| $Tap_{11-43}$ | 0.958 | 0.979 |
| $Tap_{40-56}$ | 0.958 | 0.974 |
| $Tap_{39-57}$ | 0.980 | 1.028 |
| $Tap_{9-55}$ | 0.940 | 0.980 |
| $Qc_{18}$ | 10.000 | 27.115 |
| $Qc_{25}$ | 5.900 | 16.528 |
| $Qc_{53}$ | 6.300 | 15.025 |
| $Pg_1$ | 478.635 | 332.509 |
| $Pg_2$ | 0.000 | 99.984 |
| $Pg_3$ | 40.000 | 140.000 |
| $Pg_6$ | 0.000 | 100.000 |
| $Pg_8$ | 450.000 | 263.503 |
| $Pg_9$ | 0.000 | 99.983 |
| $Pg_{12}$ | 310.000 | 237.255 |
| Cost_Pg | 51,345.000 | 48,600.060 |
| Losses | 27.835 | 22.433 |
| Emissions | 2.528 | 1.038 |

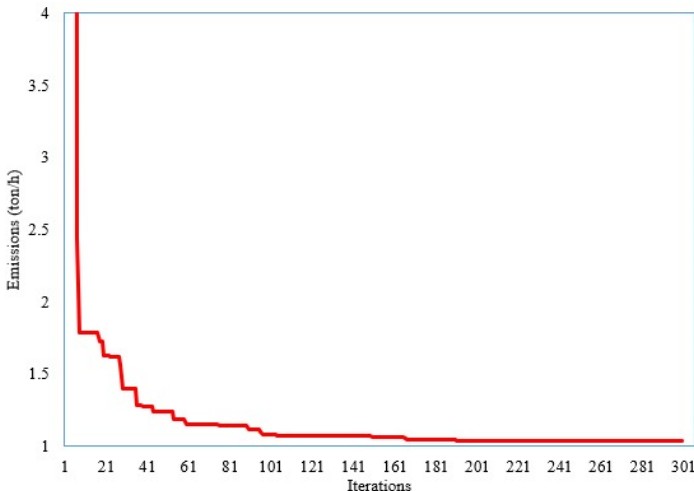

**Figure 5.** Convergence feature of the developed SNSO for Scenario 2.

**Table 5.** Comparison for Scenario 3.

| Technique | (PEs (Ton/h)) |
|---|---|
| Developed SNSO | 1.0375 |
| Social spider optimization [18] | 1.7024 |
| Enhanced social spider optimization [18] | 1.0393 |
| Teaching–learning-based optimization [37] | 1.0772 |
| Genetic algorithm [19] | 1.1210 |
| Improved genetic algorithm [19] | 1.0830 |
| Modified imperialist competitive algorithm [44] | 1.2246 |

4.1.3. OPLs Minimizing (Scenario 3)

The created SNSO achieves the minimizing of the OPLs in the fourth scenario, as shown in Table 6. In addition, Figure 6 depicts the convergence characteristic of the proposed SNSO for Scenario 3. As indicated, the gained value of OPLs is 10.195 MW, whereas the original value is 27.835 MW. This decrease is a 63.37% reduction. Table 7 shows how it compares to other metaheuristic optimization approaches. As demonstrated, the designed SNSO meets the minimal OPLs target of 10.195 MW. It surpasses the other metaheuristics of enhanced social spider optimization [18], genetic algorithm, improved genetic algorithm [19], modified differential evolution [48], salp swarm optimizer [41], and stud krill herd algorithm [49] in minimizing the OPLs.

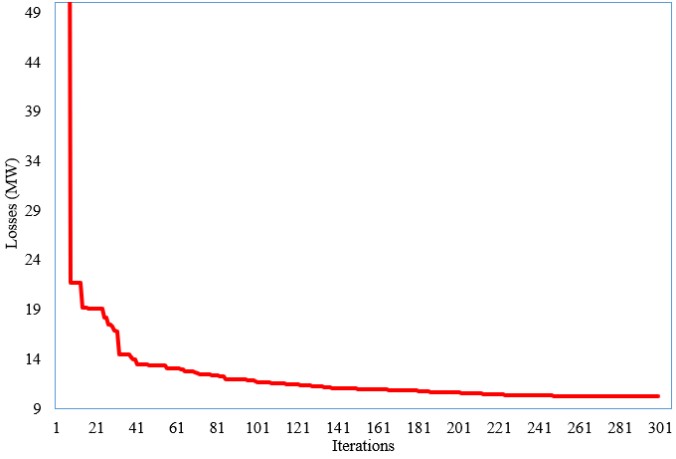

**Figure 6.** Convergence feature of the developed SNSO for Scenario 4.

**Table 6.** Optimal results using the developed SNSO for Scenario 3.

| Variables | Initial Scenario | Scenario 3 (OPLs (MW)) |
|---|---|---|
| $Vg_1$ | 1.010 | 1.021 |
| $Vg_2$ | 1.010 | 1.015 |
| $Vg_3$ | 1.010 | 1.020 |
| $Vg_6$ | 1.010 | 1.019 |
| $Vg_8$ | 1.010 | 1.024 |
| $Vg_9$ | 1.010 | 1.002 |
| $Vg_{12}$ | 1.010 | 1.007 |
| $Tap_{4-18}$ | 0.970 | 1.088 |
| $Tap_{4-18}$ | 0.978 | 0.901 |
| $Tap_{21-20}$ | 1.043 | 1.042 |
| $Tap_{24-25}$ | 1.000 | 0.981 |
| $Tap_{24-25}$ | 1.000 | 1.063 |
| $Tap_{24-26}$ | 1.043 | 0.993 |
| $Tap_{7-29}$ | 0.967 | 0.922 |
| $Tap_{34-32}$ | 0.975 | 0.961 |
| $Tap_{11-41}$ | 0.955 | 0.911 |
| $Tap_{15-45}$ | 0.955 | 0.922 |
| $Tap_{14-46}$ | 0.900 | 0.912 |
| $Tap_{10-51}$ | 0.930 | 0.915 |
| $Tap_{13-49}$ | 0.895 | 0.901 |
| $Tap_{11-43}$ | 0.958 | 0.900 |
| $Tap_{40-56}$ | 0.958 | 1.003 |
| $Tap_{39-57}$ | 0.980 | 0.991 |
| $Tap_{9-55}$ | 0.940 | 0.916 |
| $Qc_{18}$ | 10.000 | 14.634 |
| $Qc_{25}$ | 5.900 | 16.227 |
| $Qc_{53}$ | 6.300 | 13.562 |
| $Pg_1$ | 478.635 | 200.798 |
| $Pg_2$ | 0.000 | 6.558 |
| $Pg_3$ | 40.000 | 132.683 |
| $Pg_6$ | 0.000 | 99.762 |
| $Pg_8$ | 450.000 | 311.717 |
| $Pg_9$ | 0.000 | 99.949 |
| $Pg_{12}$ | 310.000 | 409.528 |
| Cost_Pg | 51,345.000 | 44,643.720 |
| Losses | 27.835 | 10.195 |

**Table 7.** Comparison for Scenario 3.

| Technique | (OPLs (MW)) |
|---|---|
| Developed SNSO | 10.1952 |
| Modified differential evolution [48] | 10.558 |
| Social spider optimization [18] | 10.614 |
| Salp swarm optimizer [41] | 11.320 |
| Modified imperialist competitive algorithm [44] | 11.883 |
| Genetic algorithm [19] | 11.814 |
| Improved genetic algorithm [19] | 10.516 |
| Stud krill herd algorithm [49] | 10.688 |

4.1.4. Assessment of the Stability of the Developed SNSO for IEEE 57-Bus Power System

The acquired objectives of the thirty runs are documented in order to perform a comprehensive assessment of the stability of the generated SNSO for all scenarios. The associated average objective is determined for each scenario, and a graph is shown to depict the proportion of each run to the average as a dynamic indicator ($Ind_{OJi}$) through

Equation (31) and, therefore, the proximity of each run to the average. Figure 7 depicts the acquired indications of the associated objective percentages via the created SNSO runs.

$$\mathrm{Ind}_{\mathrm{OJ}_i} = \frac{OJ_i}{\frac{\sum_{i=1}^{30} OJ_i}{30}}, i = 1, 2, ..m \tag{31}$$

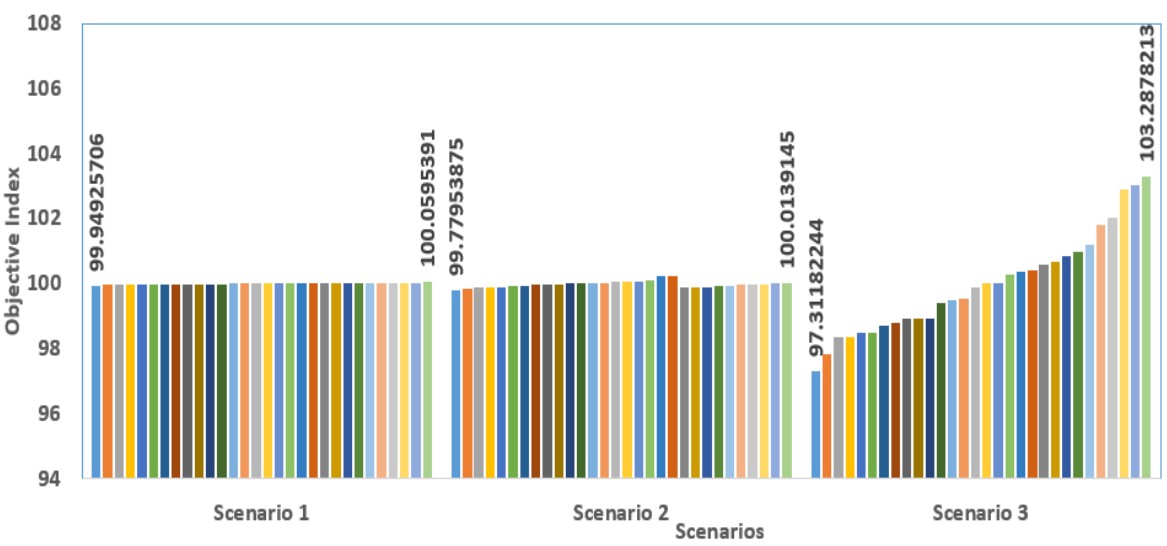

**Figure 7.** Obtained objectives percentages through the runs via the developed SNSO.

As demonstrated, the developed SNSO has the capability of always locating near percentages to 100% if its average is close to its lowest. The lowest and maximum index percentages in the first scenario are 99.949 and 100.059%, respectively, but in the second scenario, they are 99.779 and 100.014%, respectively. For the third scenario, the lowest and highest index percentages are 97.311 and 103.287%, respectively. This demonstrates the developed SNSO's exceptional stability under all circumstances.

4.1.5. Validations of Operation for Rotary and Static Machines in the IEEE 57-Bus Power System

In this section, a validation assessment has been investigated for the rotary and static machines of the IEEE 57-bus power system. To illustrate, Figure 8 plots the active power outputs of synchronous machines and the corresponding limits while Figure 9. depicts the reactive power outputs of synchronous machines and the corresponding limits for Scenarios 1–3 in the IEEE 57-bus power system. As shown, the active and reactive power outputs of all synchronous machines for all studied scenarios are within their limits with no violations. Additionally, operating Tap points and their limits of transformers for Scenarios 1–3 in the IEEE 57-bus power system are demonstrated in Figure 10. This figure illustrates that all the operating levels of the taps are inside their maximum and minimum limitations of 1.1 and 0.9, respectively. Furthermore, the reactive power outputs of VAR sources for Scenarios 1–3 in the IEEE 57-bus power system are within their limits with no violations as illustrated in Figure 11.

*4.2. The Second System Results*

The following two scenarios are investigated for the second EPS:

- Scenario 4: OV1 Minimization of FGCs
- Scenario 5: OV3 Minimization of OPLs

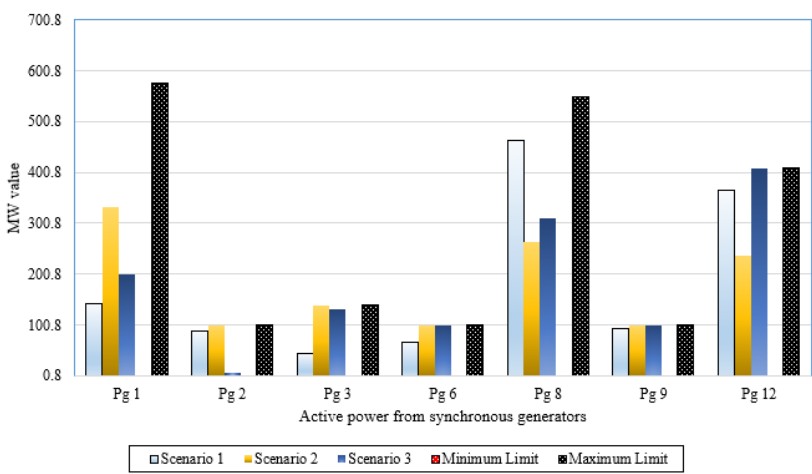

**Figure 8.** Active power outputs of synchronous machines for Scenarios 1–3 in the IEEE 57-bus power system.

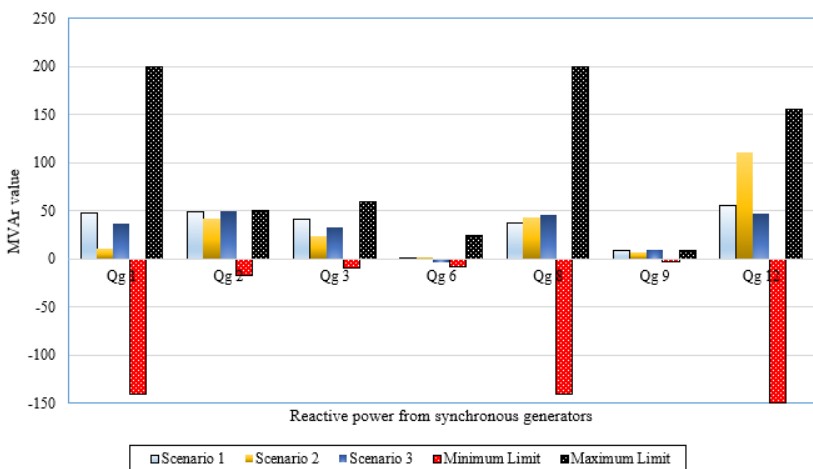

**Figure 9.** Reactive power outputs of synchronous machines for Scenarios 1–3 in the IEEE 57-bus power system.

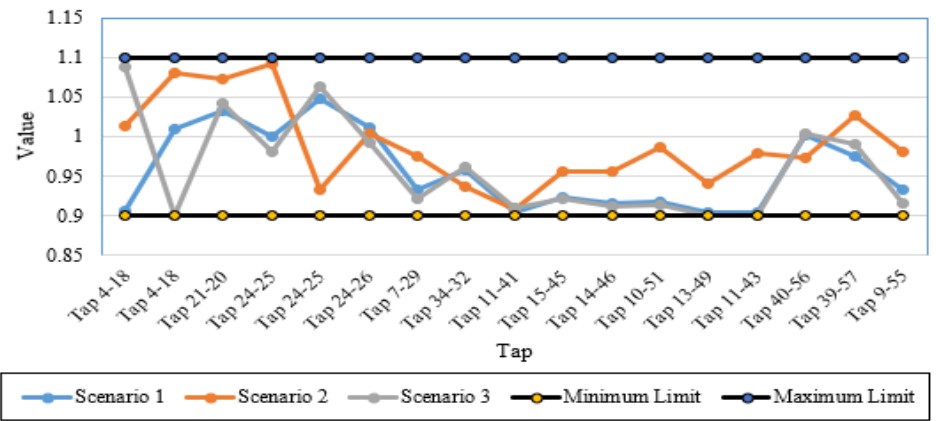

**Figure 10.** Operating Tap points and their limits of transformers for Scenarios 1–3 in the IEEE 57-bus power system.

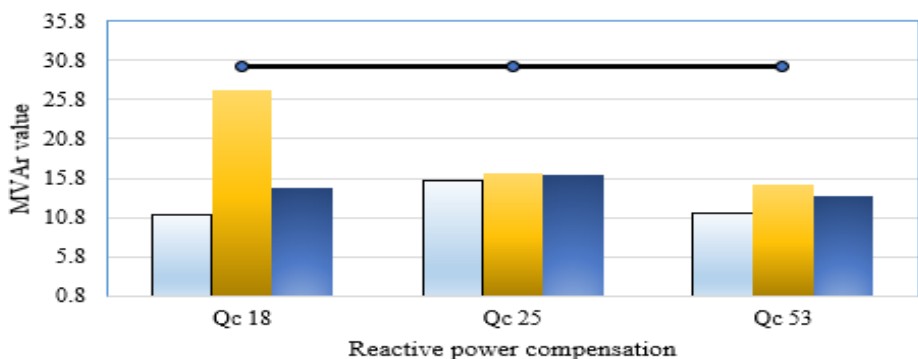

**Figure 11.** Reactive power outputs of VAR sources for Scenarios 1–3 in the IEEE 57-bus power system.

4.2.1. FGCs Minimizing (Scenario 4)

For this scenario, the developed SNSO is applied whereas their obtained outputs are recorded in Table 8. Added to that, Figure 12 illustrates the convergence feature of the developed SNSO for Scenario 4. As shown, the developed SNSO minimizes the FGCs from 25,098.70 USD/h at the initial scenario to 22,953.425 USD/h. This reduction represents a percentage of 8.54%.

**Table 8.** Optimal results using the developed SNSO for Scenario 4.

|  | Initial Scenario | Scenario 5 (FGCs (USD/h)) |
|---|---|---|
| $Vg_1$ | 1.000 | 1.060 |
| $Vg_2$ | 1.000 | 1.060 |
| $Vg_3$ | 1.000 | 1.060 |
| $Vg_4$ | 1.000 | 1.060 |
| $Vg_5$ | 1.000 | 1.060 |
| $Vg_6$ | 1.000 | 1.060 |
| $Vg_7$ | 1.000 | 1.060 |
| $Vg_8$ | 1.000 | 1.046 |
| $Pg_1$ | 85.690 | 1.052 |
| $Pg_2$ | 157.400 | 189.5608 |
| $Pg_3$ | 139.310 | 10.000 |
| $Pg_4$ | 113.690 | 214.726 |
| $Pg_5$ | 166.480 | 180.397 |
| $Pg_6$ | 31.710 | 10.000 |
| $Pg_7$ | 92.000 | 233.916 |
| $Pg_8$ | 122.490 | 56.306 |
| FGCs (USD/h) | 25,098.700 | 22,953.425 |
| OPLs (MW) | 19.015 | 37.450 |

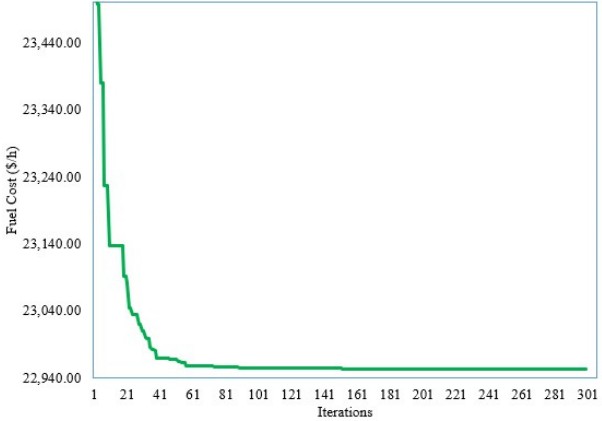

**Figure 12.** Convergence feature of developed SNSO for Scenario 4.

For this scenario, the developed SNSO is compared with several other new techniques such as enhanced grey wolf technique [50], crow search technique [35], salp swarm technique, novel bat technique, and spotted hyena technique that are applied for this scenario as tabulated in Table 9. As seen, the developed SNSO outperforms all other techniques in minimizing the FGCs where the developed SNSO obtains the least FGCs of 22,953.425 USD/h. On contrary, novel bat, salp swarm, enhanced grey wolf, spotted hyena, crow search and modified crow search techniques obtain 22,960.8, 22,965.6, 22,957.7, 22,958.8, 22,959.4 and 22,955.6 USD/h, respectively.

**Table 9.** Comparison for Scenario 5.

| Technique | FGCs (USD/h) |
|---|---|
| Novel bat technique | 22,960.810 |
| Salp swarm technique | 22,965.590 |
| Enhanced grey wolf technique | 22,957.720 |
| Spotted hyena technique | 22,958.780 |
| Crow search technique | 22,959.360 |
| Modified crow search technique | 22,955.550 |
| Developed SNSO | 22,953.425 |

4.2.2. OPLs Minimizing (Scenario 5)

For the fifth scenario, the minimization of the OPLs is obtained by the developed SNSO as reflected in Table 10. In addition, Figure 13 illustrates the convergence feature of the developed SNSO for Scenario 5. As shown, the developed SNSO minimizes the OPLs from 19.02 MW at the initial scenario to 7.24 MW. This reduction represents a percentage of 61.93%.

4.2.3. Assessment of the Stability of the Developed SNSO for WDA Power System

For the WDA power system, similar assessment methodology in Section 4.1.4, the objective indexes via Equation (31) are described for all runs of the developed SNSO. For each scenario, Figure 14 describes the obtained indicators of the related objective percentages through the runs via the developed SNSO.

**Table 10.** Optimal results using the developed SNSO for Scenario 6.

| | Initial Scenario | Scenario 6 (OPLs (MW)) |
|---|---|---|
| $Vg_1$ | 1.000 | 1.059 |
| $Vg_2$ | 1.000 | 1.060 |
| $Vg_3$ | 1.000 | 1.060 |
| $Vg_4$ | 1.000 | 1.060 |
| $Vg_5$ | 1.000 | 1.060 |
| $Vg_6$ | 1.000 | 1.060 |
| $Vg_7$ | 1.000 | 1.060 |
| $Vg_8$ | 1.000 | 1.060 |
| $Pg_1$ | 85.690 | 59.825 |
| $Pg_2$ | 157.400 | 61.159 |
| $Pg_3$ | 139.310 | 180.473 |
| $Pg_4$ | 113.690 | 129.540 |
| $Pg_5$ | 166.480 | 117.618 |
| $Pg_6$ | 31.710 | 105.142 |
| $Pg_7$ | 92.000 | 155.355 |
| $Pg_8$ | 122.490 | 87.876 |
| FGCs (USD/h) | 25,098.700 | 24,779.010 |
| OPLs (MW) | 19.015 | 7.239 |

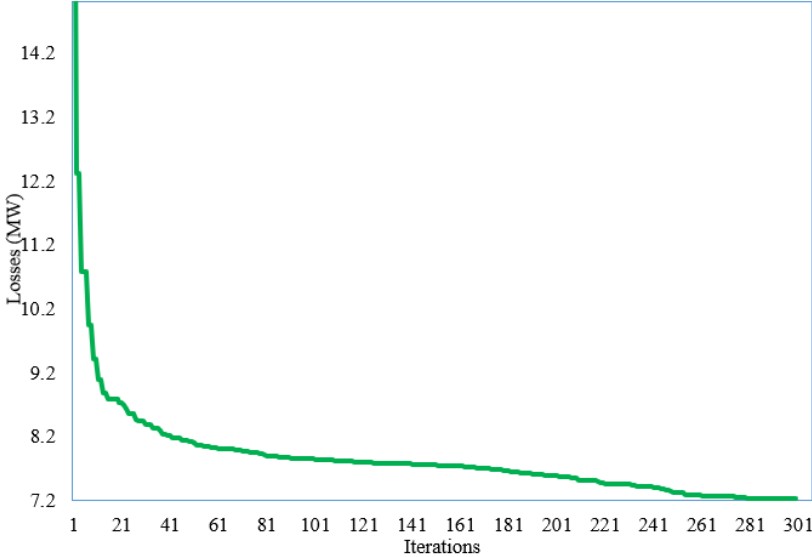

**Figure 13.** Convergence feature of the developed SNSO for Scenario 5.

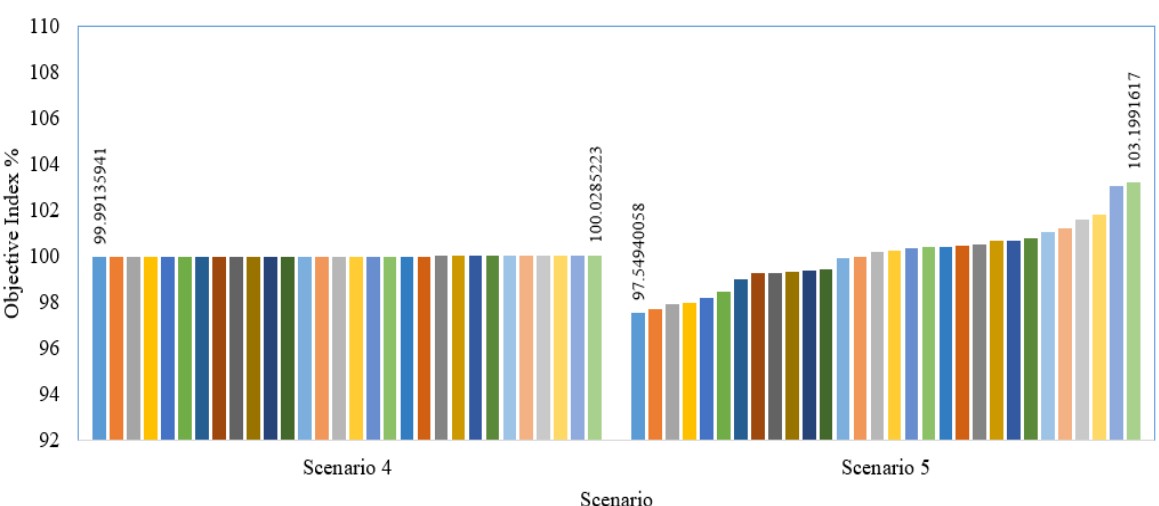

**Figure 14.** Obtained objectives percentages through the runs via the developed SNSO.

As can be shown, the evolved SNSO has the potential to always locate close percentages to 100% if its average is close to its minimum. In the first situation, the mini-mum and maximum index percentages are 99.99 and 100.03%, respectively, whereas in the second scenario, they are 97.55 and 103.2%, respectively. This displays the developed SNSO's remarkable stability in all conditions for the WDA power system.

4.2.4. Validations of Operation for Rotary Machines in the WDA Power System

In this section, a validation assessment has been conducted for the rotary machines of the WDA power system. To illustrate, the active power outputs of synchronous machines for Scenarios 4 and 5 in the WDA power system are within their limits with no violations as illustrated in Figure 15. Moreover, the reactive power outputs of synchronous machines for Scenarios 4 and 5 in the WDA power system are within their limits with no violations as illustrated in Figure 16.

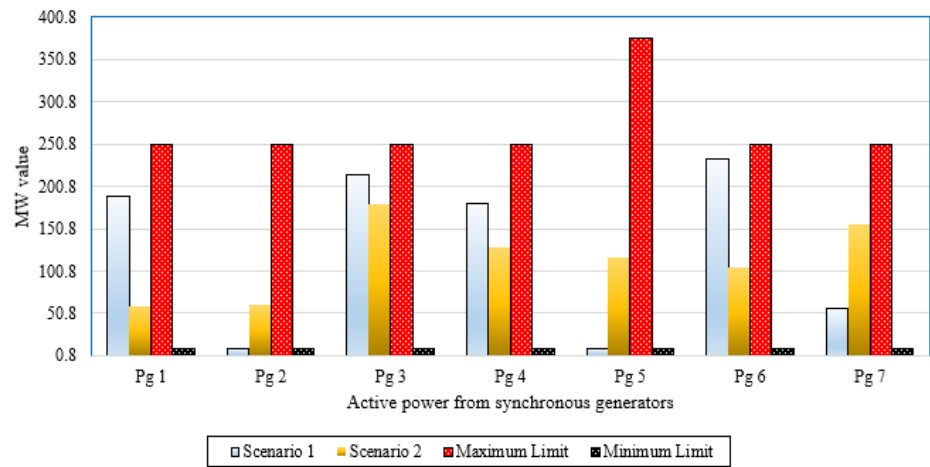

**Figure 15.** Active power outputs of synchronous machines for Scenarios 4 and 5 in the WDA power system.

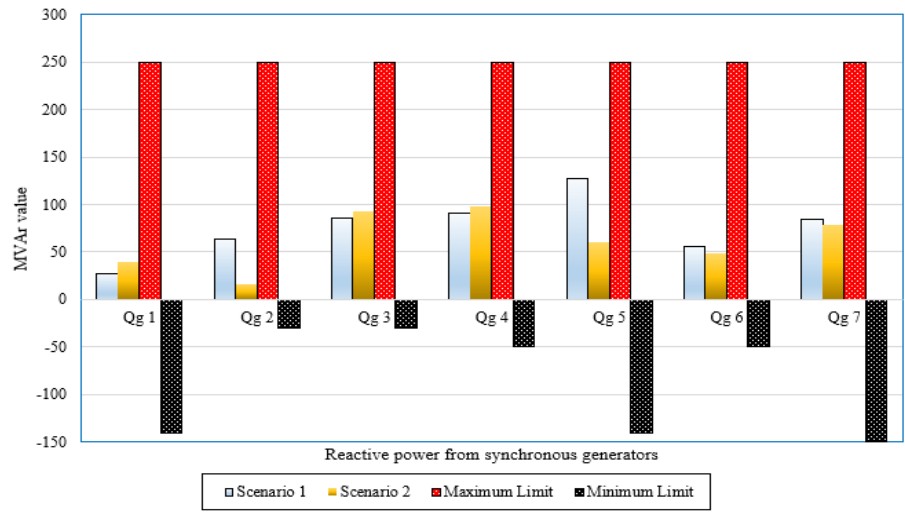

**Figure 16.** Reactive power outputs of synchronous machines for Scenarios 4 and 5 in the WDA power system.

### 4.3. Computational Burden of the Developed SNSO for Both Systems

The computation time of the designed SNSO is computed and reported in Table 11 for both systems. It is calculated as the mean time required for each iteration, incorporating the power flow technique. One such table demonstrates that the accompanying timeframe for the created SNSO is distinct for both systems, with the developed SNSO taking the shortest time by 0.545 sec in the second scenario for the IEEE 57-bus power system and 0.418 sec in the sixth scenario for the WDA power system.

**Table 11.** Computational burden of the developed SNSO for both systems.

|  |  | **Mean Time (s)/Iteration** |
| --- | --- | --- |
| IEEE 57-bus power system | Scenario 1 | 0.613 |
|  | Scenario 2 | 0.545 |
|  | Scenario 3 | 0.556 |
| WDA power system | Scenario 4 | 0.429 |
|  | Scenario 5 | 0.418 |

## 5. Conclusions

In this paper, a developed solution based on social network search (SNS) optimizer (SNSO) for optimal power system operation (OPSO) in power systems. The developed OPSO's evaluation is conducted using an IEEE standardized 57 bus power system and real Egyptian power system of the West Delta area (WDA). Five diverse scenarios are considered based on the targeted objective function of the cost of fuel, power losses, and polluting emissions. The developed SNSO derives considerable stability for all scenarios. A validation assessment is conducted for synchronous generator rotary machines and static machines of on-load tap changer (OLTC) transformers and volt-ampere reactive (VAR) sources of the IEEE 57-bus and WDA power systems which illustrate that all machines are operating inside their limits with no violation. The simulated findings prove the developed OPSO's solution accuracy and resilience when compared to other relevant techniques in the literature. Moreover, the developed SNSO declares significant effectiveness compared with various contemporary techniques. For all investigated scenarios, significant reductions are attained in the targeted goal.

- For the IEEE standardized 57 bus power system, The reduction percentage is reached to represent a percentage of 18, 14.39% and 63.37% for scenarios 1–3 compared to the initial scenario, respectively.
- For the real Egyptian power system of WDA, the reduction percentage is reached to represent a percentage of 8.54% and 61.95% for scenarios 4 and 5 compared to the initial scenario, respectively.

In contrast, the reliability of the developed SNSO requires more support and more applications in the area of power system optimization should be verified including renewable sources. Therefore, it is recommended as future work to derive enhanced versions of the SNSO and apply it for mathematical benchmark models and engineering optimization problems in the field of power systems.

**Author Contributions:** Conceptualization, R.E.-S. and A.S. methodology. A.S. and R.E.-S.; software and validation, formal analysis, investigation, A.G.; resources, A.E. data curation, writing—original draft preparation, E.E.; writing—review and editing, R.E.-S.: supervision, E.E.; funding acquisition. All authors have read and agreed to the published version of the manuscript.

**Funding:** This research was funded Taif University Researchers Supporting Project number (TURSP-2020/86), Taif University, Taif, Saudi Arabia.

**Institutional Review Board Statement:** Not applicable.

**Informed Consent Statement:** Not applicable.

**Acknowledgments:** This work was supported by Taif University Researchers Supporting Project number (TURSP-2020/86), Taif University, Taif, Saudi Arabia.

**Conflicts of Interest:** The authors declare no conflict of interest.

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
