# Peer review of "Scheduling of Generation Stations, OLTC Substation Transformers and VAR Sources for Sustainable Power System Operation Using SNS Optimizer"

_sustainability, doi:10.3390/su132111947_

Round 1
Reviewer 1 Report
The authors present an approach for scheduling of generation stations, OLTC substation transfromers and VAR sources using SNS Optimizer.
- In the introduction the authors presented state of the art literature, I suggest authors to comment of shortcomings of earlier literature and how this paper is going to overcome those shortcomes.
- Comment on the impact of renewable energy sources on the proposed algorithm as forecasting renewable energy can be challening.
- Improve figure quality
- Comment on time taken to solve the problem and comment of size of system vs time taken
- Apart from using SNS optimizer, comment of significant contributions of this paper compared to earlier papers in literature.
Author Response
The response is attached.

Reviewer 2 Report
The authors proposed a scheduling of OLTC and SVCs for OPSO using a recent algorithm of social network search (SNS) optimizer. The OPSO is performed by achieving many optimization targets of targets of cost of fuel, power losses, and polluting emissions. The SNS optimizer is developed for handling the OPSO problem and applied on an IEEE standardized 57-bus power system and real Egyptian power system of West Delta Area. The developed SNS optimizer was used in various assessments and quantitative analysis with various contemporary techniques. The content of this manuscript is important, references are relevant recently and the manuscript is well-written. However, this paper needs more explanation. To really check the specificity and effectiveness of the proposed strategy, I think the authors need to clarify the following:
- What’s the meaning of ai, bi, ci, ??? The objective function is cost, but there seems to be no definition of cost.
- It seems necessary to explain the concept using illustration to explain more clearly.
- How much time was simulated with the schedule? How many periods?
- Looking at the results in Table 2, other methods derive more optimal values. Is the method in this paper the optimal value?
- It looks like authors needs to show detailed simulation results.
Author Response
Response to reviewer # 2 is attached.

Reviewer 3 Report
The English expressions and style should be corrected.
The indices in the equations should be checked again and corrected where necessary.
The explanations concerning the moods could be improved. Comments should be added concerning which mode parameters are used as elements of the vectors described.
I attach a file with sticky notes with some errors to be corrected.
The optimal tap values are very strange. I do not know such devices. Usually they operate with some steps in positive and negative direction.
The comparisons with other heuristic methods given in tables 2, 4, 6, and 8 are not correct. They could not be compared only on the bases of fuel costs or losses or emissions reduction but the whole solutions should be revealed and analyzed, or at least there should be comments concerning the results for the other quantities (mode/regime parameters). Some (I recommend sufficiently extensive) comments about realization of this other methods should be included.
In fig.8 the min active power is higher than the obtained optimal values. It is not possible in fact... The maximal power is not shown at all.

Author Response
Response to reviewer # 3 is attached.

Reviewer 4 Report
This paper uses the latest social network search optimization algorithm to coordinate machine scheduling in order to realize the optimal operation of power system. The proposed approach is good but some important issues should be explained, to improve it further.
1. This work reviews many papers, but lack of tight connection among them. The authors should really improve their literature review a lot.
2. For equations (14-21), we need better interpretations in your REVISION. Please do it and make it as clear as possible.
3. Where are the data sources from Tables 1 to 10? It is preferred to have real data.
4. The shortcomings of the proposed method are suggested to be included in “Conclusions”.
5. The whole article needs further polishing to correct grammar errors and typos. Particularly in Sect 4. It is now NOT clear to common readers. Please could you re-write this whole section and make sure what you said is sufficiently clear to most common readers in this journal.
Author Response
Response to Reviewer 4
This paper uses the latest social network search optimization algorithm to coordinate machine scheduling in order to realize the optimal operation of power system. The proposed approach is good but some important issues should be explained, to improve it further.
Response: The authors are thankful to the reviewer for his important notes. The authors hope that the revised version meets his satisfaction.
- This work reviews many papers, but lack of tight connection among them. The authors should really improve their literature review a lot.
Response: The introduction section is revised and corrected where comments of shortcomings of earlier literature are added, and the main contributions of the paper are illustrated.
- For equations (14-21), we need better interpretations in your REVISION. Please do it and make it as clear as possible.
Response: Ok. better interpretations to equations (14-21) are provided in the revised manuscript for more clarifications.
- Where are the data sources from Tables 1 to 10? It is preferred to have real data.
Response: Ok. the data sources of all tables are revised and cited:
Table 1 is data that is taken from ref. [67].
Table 2 contains the results of the proposed method and results published from Enhanced Social Spider Optimizer [46], Slime mould algorithm [79], Salp Swarm Optimizer [74] and Improved Salp Swarm Optimizer [78]
Table 3 contains the results of the proposed method and results published from Real Coded Biogeography-Based Optimization [30], Social Spider Optimization [46], Enhanced Social Spider Optimization [46], Bat Search Algorithm [72], differential search algorithm [77], Electromagnetic Field Optimization [73], Salp Swarm Optimizer [74], Modified Imperialist Competitive Algorithm [75], Cuckoo Optimization [76], Genetic Algorithm [50], Improved Genetic Algorithm [50] and Improved Salp Swarm Optimizer [78]
Table 4 contains the results of the proposed method.
Table 5 contains the results of the proposed method and results published from Social Spider Optimization [46], Enhanced Social Spider Optimization [46], Teaching-Learning Based Optimization [68], Genetic Algorithm [50], Improved Genetic Algorithm [50] and Modified Imperialist Competitive Algorithm [75]
Table 6 contains the results of the proposed method.
Table 7 contains the results of the proposed method and results published from Modified Differential Evolution [80], Social Spider Optimization [46], Salp Swarm Optimizer [74], Modified Imperialist Competitive Algorithm [75], Genetic Algorithm [50] , Improved Genetic Algorithm [50] and Stud Krill Herd Algorithm [81]
Tables 8:11: contains the simulation results of the proposed method and different techniques.
- The shortcomings of the proposed method are suggested to be included in “Conclusions”.
Response: Ok. The conclusion section is updated to show the several merits of the proposed methodology. As well, the shortcomings are stated to be treated as a future work.
- The whole article needs further polishing to correct grammar errors and typos. Particularly in Sect 4. It is now NOT clear to common readers. Please could you re-write this whole section and make sure what you said is sufficiently clear to most common readers in this journal.
Response: Ok. The whole article is revised, and the grammar errors and typos are corrected. As well, the English expressions and style are revised and corrected. Further, Section 4 is re-written for more clarifications to the readers.

Round 2
Reviewer 1 Report
Authors addressed all my comments
Author Response
Thank you for your positive feedback about our paper.
Reviewer 2 Report
The manuscript was well edited.
Author Response

(The authors gave the same response as above.)

Reviewer 3 Report
The present version of the article is improved, but not all of my concerns were sufficiently addressed.
Author Response
Comment: The present version of the article is improved, but not all of my concerns were sufficiently addressed.
Response: Thank you for your positive response about our paper. All raised comments are revisited again to improve the previous response.
Reviewer 4 Report
The revised paper can be accepted in its current form.
Author Response
Thanks for positive response about our revised version